# Roles of Nrf2 in Protecting the Kidney from Oxidative Damage

**DOI:** 10.3390/ijms21082951

**Published:** 2020-04-22

**Authors:** Masahiro Nezu, Norio Suzuki

**Affiliations:** 1Department of Endocrinology and Diabetes, Yamanashi Prefectural Central Hospital, Fujimi 1-1-1, Kofu, Japan; nezumasahiro@med.tohoku.ac.jp; 2Division of Oxygen Biology, United Centers for Advanced Research and Translational Medicine, Tohoku University Graduate School of Medicine, Seiryo-machi 2-1, Aoba-ku, Sendai, Japan

**Keywords:** ischemia-reperfusion injury, chronic kidney disease, Nrf2-activating compounds, erythropoietin, kidney fibrosis

## Abstract

Over 10% of the global population suffers from kidney disease. However, only kidney replacement therapies, which burden medical expenses, are currently effective in treating kidney disease. Therefore, elucidating the complicated molecular pathology of kidney disease is an urgent priority for developing innovative therapeutics for kidney disease. Recent studies demonstrated that intertwined renal vasculature often causes ischemia-reperfusion injury (IRI), which generates oxidative stress, and that the accumulation of oxidative stress is a common pathway underlying various types of kidney disease. We reported that activating the antioxidative transcription factor Nrf2 in renal tubules in mice with renal IRI effectively mitigates tubular damage and interstitial fibrosis by inducing the expression of genes related to cytoprotection against oxidative stress. Additionally, since the kidney performs multiple functions beyond blood purification, renoprotection by Nrf2 activation is anticipated to lead to various benefits. Indeed, our experiments indicated the possibility that Nrf2 activation mitigates anemia, which is caused by impaired production of the erythroid growth factor erythropoietin from injured kidneys, and moderates organ damage worsened by anemic hypoxia. Clinical trials investigating Nrf2-activating compounds in kidney disease patients are ongoing, and beneficial effects are being obtained. Thus, Nrf2 activators are expected to emerge as first-in-class innovative medicine for kidney disease treatment.

## 1. Introduction

The kidneys are two organs located in the retroperitoneal space and function not only to purify blood but also to control blood pressure by producing the vasopressor renin, to maintain bone mineral homeostasis by activating vitamin D, and to produce the erythroid growth factor erythropoietin (Epo). Therefore, kidney injuries provoke fluid-balance abnormalities, uremia, hypertension, born-mineral disorders and anemia [1]. Although more than 850 million people worldwide suffer from kidney disease, which carries a considerable burden of medical expenses in many countries [2], no effective medicine for treating kidney disease has been established due to the lack of knowledge of the molecular pathology of multifactorial renal failure. Thus, elucidation of the molecular mechanisms of kidney disease progression is urgently needed.

Among the different types of renal failure, acute kidney injury (AKI) [3] and chronic kidney disease (CKD) are the major syndromes [4]. AKI is defined as an abrupt reduction in renal function within a few days or weeks [5], while CKD is defined by the presence of kidney damage or decreased kidney function for three or more months [6]. Restoration of kidney structure and recovery of renal function are often observed in AKI patients [7], even though AKI is associated with poor mortality [8]. Patients suffering from AKI bear a heightened risk of disease progression to CKD followed by end-stage renal disease (ESRD), which is the final common clinical feature of renal failure [9]. For patients with ESRD, renal replacement therapies, including hemodialysis, peritoneal dialysis and kidney transplantation, are fundamentally unavoidable [10]. On the other hand, considering that AKI and CKD interact with each other, it is known that patients with CKD bear a higher risk of developing AKI [11].

For kidney functions such as reabsorption and humoral factor secretion, kidneys are rich in capillaries compared with other organs. Renal capillaries are considered to be easily damaged by hyperglycemia and hypertension, the major causes of kidney disease, and capillary damage provokes IRI [12]. Since IRI generates oxidative stress, which is involved in various diseases, it is plausible that a reduction in oxidative stress could block kidney disease progression. Indeed, activators of Nrf2 (nuclear factor erythroid 2-related factor 2; a member of the Cap ‘n’ Collar [CNC] transcription factor family), which is the master regulator of antioxidant genes [13,14], are investigated in ongoing clinical trials as medicines for treating kidney disease [15,16,17]. The Nrf2 activators currently in development are compounds that inhibit Keap1 (Kelch-like ECH-associated protein 1), which negatively regulates Nrf2 under conditions free from oxidative stress [14,18]. In this review, we introduce the molecular functions of Keap1 and Nrf2 in kidney disease as well as up-to-date information on the development of drugs targeting the Keap1–Nrf2 system for kidney disease treatment.

## 2. Oxidative Stress in Kidney

Renal tubular epithelial cells actively and constitutively produce adenosine triphosphate for reabsorption of water and solutes from pre-urine, consuming oxygen by the abundant mitochondria [19]. Mitochondrial respiration also produces reactive oxygen species, which are highly reactive to biomolecules, including genomic DNA. Low-level reactive oxygen species (ROS) are essential for intra- and intercellular signaling to maintain kidney homeostasis and function, including vascular reactivity, renal hemodynamics, glomerular filtration, tubular reabsorption and hormonal secretion, while excessive ROS are detrimental to renal cells by leading to oxidative stress [20].

To deliver oxygen into the kidneys via blood flow, approximately 20–25% of the cardiac output is allocated to the kidneys, which is significantly high considering the size of the kidneys compared to that of other organs [21]. However, the characteristics of the renal vasculature forming many arterial-to-venous shunts often cause irregular blood flow and IRI [21]. In the reperfusion phase of renal IRI, ROS are produced by the mitochondrial respiration chain and/or nicotinamide adenine dinucleotide phosphate (NADPH) oxidases [3,21,22]. In addition to ROS, the accumulation of electrophilic molecules triggers oxidative stress. Interestingly, an electrophilic molecule, 15-deoxy-∆12,14)-prostaglandin J2, is produced by convoluted blood flow with shear stress and activates Nrf2 in endothelial cells [23].

Excessive ROS and electrophiles impair cellular homeostasis and function as oxidative stress followed by cell death, which leads to inflammation, tissue damage and fibrosis (Figure 1) [14,20]. Inflammatory cells, which are locally activated in the renal microenvironment or migrate from hematopoietic organs, further produce ROS. Thus, oxidative stress, as with inflammation, is regarded to be a key aggravating factor for the initiation and progression of AKI and CKD, including diabetic nephropathy, hypertension-associated kidney disease and toxic-induced nephropathy [18,20,24]. In these types of kidney disease, electrophiles accumulate as byproducts of impaired detoxification and abnormal metabolism and aggravate oxidative stress, making matters worse [25,26]. Therefore, defense against oxidative stress is regarded as an important therapeutic target for preventing kidney disease progression.

## 3. Role of the Keap1-Nrf2 System in Cytoprotection against Oxidative Stress

To protect cells from harmful oxidative stress, the transcription factor Nrf2 plays an integrative role in inducing the expression of genes encoding enzymes involved in antioxidant (e.g., glutathione and NADPH) production and the reduction in pro-oxidants (e.g., heme and quinonoids, Figure 2) [14,18]. In unstressed cells, Nrf2 is constitutively synthesized but degraded by the Nrf2-specific ubiquitin ligase complex, in which the stress-sensor molecule Keap1 directly captures Nrf2 (Figure 2). A Keap1 molecule contains reactive cysteines, which are adducted by oxidants and electrophiles, to sense cellular oxidative stress. Since cysteine-modified Keap1 no longer binds to Nrf2, Nrf2 avoids degradation and activates the expression of its target genes under oxidative stress conditions [14,18].

Several proteins have been identified as essential transcriptional cofactors directly binding to Nrf2 in nuclei. The heterodimer formation of Nrf2 with one of the small Maf (musculoaponeurotic fibrosarcoma) proteins (MafF, MafG, and MafK) is necessary for Nrf2 binding to CsMBE (CNC and small Maf binding element, 5′-(^A^/_G_)TGA(^G^/_C_)nnnGC) located in promoters or enhancers of Nrf2-target genes (Figure 2) [27]. In addition to DNA-binding factors, such as small Maf proteins, cyclic AMP-response element binding protein (CREB)-binding protein (CBP), brahma-related gene 1 (BRG1), and chromodomain helicase DNA binding protein 6 (CHD6) are involved in transcriptional activation of genes related to cytoprotection against oxidative stress by directly binding to Nrf2 (Figure 2) [27,28,29,30]. Additionally, we previously discovered that the transcriptional mediator complex contributes to antioxidant gene expression via a direct interaction between Nrf2 and a mediator subunit, MED16 (Figure 2) [31].

## 4. Mouse Genetic Models Used to Investigate the Roles of Nrf2 in the Kidney

Systemic Nrf2 knockout (Nrf2-KO) mice exhibit no abnormalities at birth, during growth, with fertility or throughout their lifespan under normal conditions [32], whereas analyses of Nrf2-KO mice with various disease models have demonstrated that Nrf2 plays protective roles against respiratory disease [33,34,35,36], cardiac disease [37], neurological disease [38,39], diabetes mellitus [40,41,42,43], obesity [44], sickle-cell disease [45], inflammation [46,47], infection [48], liver disease [49,50,51,52] and sensory organ diseases [53,54,55]. Additionally, in kidney disease models such as autoimmune nephritis [56], diabetic nephropathy [57,58,59], toxic injury [60,61,62], ureteral obstruction [63], podocyte injury [64] and IRI [61,63,64,65,66,67], the loss of Nrf2 aggravates tissue damage and fibrosis (Table 1).

On the other hand, systemic Keap1 knockout (Keap1-KO) mice starve to death during the weaning period because of impaired feeding due to hyperkeratosis of the upper gastrointestinal tract, suggesting that Nrf2 induces the expression of genes encoding keratin [68]. Indeed, genetic deletion of Nrf2 in keratinocytes rescues Keap1-KO mice from lethal hyperkeratosis [69]. However, most of the rescued mice die within 10 months after birth due to polyuria and dehydration caused by an abnormal renal structure resembling hydronephrosis. *Keap1* gene deletion exclusively in renal tubular epithelial cells of mouse embryos also induced polyuria and hydronephrosis after birth [69,70]. These results indicate that constitutive activation of Nrf2 in renal tubules of developing kidneys hinders the development and function of the kidneys (Table 1). In contrast, adult-onset deletion of the *Keap1* gene exclusively in renal tubular cells caused no abnormalities in renal structure or function [67], demonstrating that overactivation of Nrf2 in the mature renal epithelium makes kidneys resistant to oxidative injuries without any significant abnormalities. These results are consistent with the results from Nrf2-KO mice, which suggest that Nrf2 is necessary for renoprotection from injuries as described above (Table 1).

## 5. Mechanism of Renoprotection by Nrf2

Our mouse analyses have demonstrated that 8-hydroxydeoxyguanosine (8-OHdG), one of the oxidative DNA adducts and a marker for oxidative injury, accumulates in renal tubules of the outer medulla 24 h after renal IRI surgery [67]. The outer medullary areas positive for 8-OHdG, in which necrosis and apoptosis are also observed, expand to the renal cortex for seven days after injury, followed by interstitial fibrosis. These observations suggest that oxidative stress is one of the triggers for kidney disease and that quenching oxidative stress by Nrf2 activation in tubules is a plausible strategy for preventing kidney damage. In fact, systemic activation of Nrf2 in Keap1 hypomorphic mutant (Keap1 knockdown; Keap1-KD) mice significantly attenuates 8-OHdG accumulation and mitigates kidney damage progression compared to wild-type mice (Table 1) [67]. On the other hand, the Nrf2-KO mice exhibited more severe 8-OHdG accumulation and more severe tubular damage than the wild-type mice (Table 1).

Oxidative stress provoked by IRI immediately and slightly induces the expression of Nrf2-target genes in wild-type kidneys, but the induction is transient and returns to basal levels within 24 h after injury [67]. Therefore, it is suggested that Nrf2 is swiftly activated in response to the oxidative stress induced by renal IRI, but the activation is insufficient to eliminate cytotoxic stress. Additionally, the oxidative stress response system governed by Nrf2 is likely impaired due to cellular oxidative damage, as previous articles reported that Nrf2 is inactivated in chronically damaged kidneys [73]. In accordance with the theory of the Keap1-Nrf2 system, Keap1-KD mice showed significantly mild pathology of tubular damage, interstitial fibrosis and renal dysfunction (increased serum creatinine level) compared to wild-type mice (Table 1 and Figure 3A) [67].

The renoprotective effect of Nrf2 activation in systemic Keap1-KD mice is nicely recapitulated in a gene-modified mouse line in which the *Keap1* gene is deleted exclusively in adult renal tubular cells (Table 1 and Figure 3A) [67]. However, Keap1 deletion in mouse myeloid cells, which accumulate in the tubular interstitium of injured kidneys, shows no protective effect on kidney damage. Thus, tubular epithelial cells, but not myeloid cells, are responsible for the Nrf2-mediated renoprotection against oxidative stress. T lymphocytes also accumulate in the interstitia of injured kidneys in response to renal oxidative stress and participate in renal inflammation (Table 1) [66]. Intriguingly, since deletion of the *Keap1* gene in mouse T lymphocytes mitigates kidney damage caused by IRI, T lymphocytes are also suggested to be involved in Nrf2-mediated renoprotection from oxidative stress [66].

In tumor cells, Nrf2 plays an important role in cell proliferation by activating the pentose phosphate pathway (PPP), which provides nucleotides for genomic replication of mitotic tumor cells and the antioxidant NADPH for cytoprotection from oxidative stress provoked by the tumor environment [74]. Consistently, both the NADPH concentrations and the expression levels of PPP genes such as *G6PD* and *PGD* (encoding glucose-6-phosphate dehydrogenase and 6-phosphogluconate dehydrogenase, respectively) in Keap1-KD kidneys are higher than those in wild-type kidneys, and the levels are further elevated by IRI [67]. However, no differences in renal cell proliferation after IRI were observed among the Keap1-KD, Nrf2-KO and wild-type mice. Therefore, we propose that PPP activation by Nrf2 is associated with renoprotection from oxidative injury, but not cell proliferation, by producing the antioxidant NADPH. Nrf2 also induces glutathione-related genes, including *GSTM1* and *GCLM* (encoding glutathione S-transferase µ1 and glutamate-cysteine ligase modifier subunit, respectively), and matrix-assisted laser desorption/ionization-imaging mass spectrometry has demonstrated that compared to wild-type mice, glutathione is highly produced in the cortex and outer medulla in Keap1-KD mice before and after IRI [67]. Because the tubules in the renal cortex and outer medulla are vulnerable to oxidative injury (Figure 3A), glutathione production in these tissues is considered significant for kidney protection from IRI. Consequently, enhancing the production of antioxidants such as NADPH and glutathione in tubular cells is considered the major mechanism of Nrf2-mediated renoprotection from IRI (Figure 4A).

## 6. Beneficial Effects of Nrf2 Activation for Kidney Disease Treatment

Since numerous mouse genetic studies have demonstrated the beneficial effects of Nrf2 activation on the prevention of kidney disease (Table 1), basic research and clinical trials of chemical compounds activating Nrf2 are being conducted (Table 2). Above all, 1-[2-cyano-3-,12-dioxooleana-1,9(11)-dien-28-oyl]-imidazolide (CDDO-imidazolide), which is a semisynthetic triterpenoid from natural products that inhibits association between Nrf2 and Keap1 by directly binding to Keap1, is one of the most predominant compounds activating Nrf2 [75,76,77]. Oral administration of CDDO-imidazolide every two days after IRI dramatically mitigates kidney damage in mice (Figure 3B) [67]. Importantly, our data from mouse experiments propose that immediate (within 24 h) initiation of treatment with CDDO-imidazolide is essential for protection of the kidneys from oxidative damage (Figure 3B) [67].

While most reports concerning Nrf2 function in the kidney highlight beneficial effects of Nrf2 activation, several reports note its negative roles. In mouse models of autoimmune nephritis and diabetic nephropathy, Nrf2 deficiency ameliorates the disease phenotype (Table 1) [71,72]. Additionally, pharmacological Nrf2 activation worsens diabetic nephropathy in rats [78]. Although the use of Nrf2 activators in clinical trials successfully improved the estimated glomerular filtration rate (eGFR) in patients with diabetic nephropathy, increased albuminuria was observed [16]. Moreover, Nrf2 activation has been found in kidney cancers and is considered involved in cancer progression [79,80,81]. Thus, Nrf2 activation does not always lead to positive effects on kidney diseases.

In addition to urine production, the kidney produces and secretes the erythroid growth factor Epo in interstitial fibroblasts referred to as “REP (renal Epo-producing) cells” [103]. Therefore, kidney injury often causes anemia as a complication of CKD through inactivation of renal Epo production (Figure 1) [104,105]. Recent reports have suggested that Epo-deficiency anemia further aggravates kidney damage [106,107]. In fact, our mouse model for Epo-deficiency anemia [108] demonstrates that tubular damage and fibrosis are more severe than those in the control mice after unilateral renal IRI (Figure 4B and C). Although the aggravating mechanism is still controversial, enhanced oxidative stress resulting from reduced oxygen delivery (hypoxia) due to anemia, IRI due to sparse erythrocytes in renal capillaries, and/or retardation of iron utilization for hemoglobin synthesis are considered the major causes (Figure 1). Additionally, loss of Epo, which may directly protect renal cells from injury [109,110], may devastate renal damage in Epo-deficient anemic mice.

We found that CDDO-imidazolide enhances the anemia-induced activity of *Epo*-gene expression in the kidneys from anemic mice (Figure 4D). Since enhancement is not observed in normal kidneys, it is proposed that renal *Epo* gene expression is not fully induced by anemia due to anemia-provoked oxidative stress. Additionally, we propose that Nrf2 activation potentially mitigates Epo deficiency in injured kidneys, which is led by oxidative stress, although additional experiments are required to confirm this conclusion. Taken together, Nrf2 activators are thought to protect kidneys from oxidative damage via induction of antioxidant genes and prevention of Epo deficiency by intervening in the negative spiral consisting of oxidative stress and tubular damage (Figure 4A).

## 7. Clinical Impact of Pharmacological Activation of Nrf2 on Kidney Injury

Based on cumulative evidence of beneficial effects of Nrf2 activators on kidney disease treatment, CDDO-methyl ester (CDDO-ME, also known as RTA402 or bardoxolone methyl; Table 2) has been undergoing clinical trials for the treatment of CKD with type 2 diabetes mellitus [15,16]. At first, in 2006, CDDO-ME was expected to be an antitumor agent and was tested in a phase I clinical trial for solid cancer and hematological malignancy [112]. Unexpectedly, the trial demonstrated that CDDO-ME dramatically (26%) increased the eGFR of subjects suffering from kidney disease as a complication. Since then, CDDO-ME has been investigated to develop novel medicines against diabetic nephropathy, for which no effective medicine has been established despite its high morbidity [113,114]. The phase II trials of CDDO-ME, one of which has also been known as the BEAM study, confirmed that CDDO-ME dramatically improves eGFR (Table 3) [15].

Then, a phase III clinical trial (referred to as the BEACON study) was conducted with 2185 patients belonging to the second most severe CKD stage (Grade 4) [115]. Although the BEACON study showed a significant increase in eGFR by 5.5 mL/min/1.73 m^2^ in the group treated with 20 mg/day CDDO-ME, this clinical trial was terminated in 2012 because of the high incidence of cardiovascular events, especially heart failure, in the group treated with CDDO-ME (Table 3) [16]. Most of the CDDO-ME subjects also exhibited fluid retention with an increase in blood pressure, heart rate, and serum brain natriuretic peptide (BNP) in addition to a decrease in blood concentrations of albumin and hemoglobin compared to the placebo subjects. Accordingly, it has been considered that the cardiac function of patients with very poor renal function is likely vulnerable to fluid overload, which may be related to CDDO-ME administration [116]. Indeed, no imbalance in cardiovascular events was observed between the CDDO-ME and placebo subjects in the BEAM study, in which the subjects suffered from milder stages of CKD (Grade 3b and 4) than those in the BEACON study (Grade 4). Moreover, basal serum BNP concentrations, which are well correlated with fluid retention, in the CDDO-ME subjects were less than 200 pg/mL, and the incidence of cardiac failure among these subjects was comparable to that among the placebo subjects in the BEACON study [116].

In 2015, a phase II randomized double-blind, placebo-controlled clinical trial of CDDO-ME was restarted with 120 CKD patients of Grade 3 or 4, excluding patients bearing risk factors for fluid overload (e.g., high-level BNP and history of heart failure), and the results nicely showed a significant increase in eGFR without any signs of fluid overload (TSUBAKI study; Table 3) [17,18]. In 2018, a phase III randomized controlled clinical trial referred to as the AYAME study (https://clinicaltrials.gov/ct2/show/NCT03550443) was launched with diabetic CKD patients whose CKD stages are Grade 3 or 4 as a successor to the phase II study in Japan. The primary endpoint is the time to onset of a more than 30% decrease in eGFR from baseline. The AYAME study will be completed in 2022, and the data are expected to clearly determine the clinical benefits of CDDO-ME in patients suffering from diabetic nephropathy [117].

In addition to diabetic nephropathy, clinical trials of CDDO-ME for the treatment of various other types of kidney disease are currently being conducted or prepared. For example, the CARDINAL study (phase II/III, 2017–2020, https://clinicaltrials.gov/ct2/show/NCT03019185) and the FALCON study (phase III, 2019–2023, https://clinicaltrials.gov/ct2/show/NCT03918447) are ongoing with patients suffering from Alport syndrome (a hereditary progressive renal disease caused by abnormal type IV collagen) and ADPKD (autosomal dominant polycystic kidney disease), respectively [118,119]. Additionally, the long-term safety and tolerability of CDDO-ME for CKD patients are being estimated by the EAGLE study (phase III, 2019–2024, https://clinicaltrials.gov/ct2/show/NCT03749447). These clinical trials are based on the profound function of Nrf2 in renoprotection, and reducing oxidative stress by Nrf2 activation is expected to mitigate a variety of diseases beyond kidney disease.

## 8. Conclusions and Perspective

Currently, approximately 10% of the worldwide population suffers from kidney disease, and the burden of medical expenses due to kidney disease is increasing in many countries because only transplantation or hemodialysis are available for treating kidney disease. Therefore, CDDO-ME and other Nrf2 activators are earnestly anticipated as first-in-class medicine for kidney disease treatment, and the clinical trials introduced above are receiving considerable attention. Basic studies of the Nrf2 pathway have demonstrated multiple cytoprotective functions of Nrf2 and unveiled regulatory mechanisms of Nrf2 activity in response to oxidative stress. Recent studies have shown that activated Nrf2 in nuclei is degraded through βTrCP-mediated ubiquitination to avoid overactivation of Nrf2 [90,99,120], supporting the safety of chemical compounds activating Nrf2 for patients. Because oxidative stress is a primary factor for tissue damage in every organ, pharmacologically driving the antioxidative response system by Nrf2 activators is plausible for curing a variety of diseases.

## Figures and Tables

**Figure 1 ijms-21-02951-f001:**
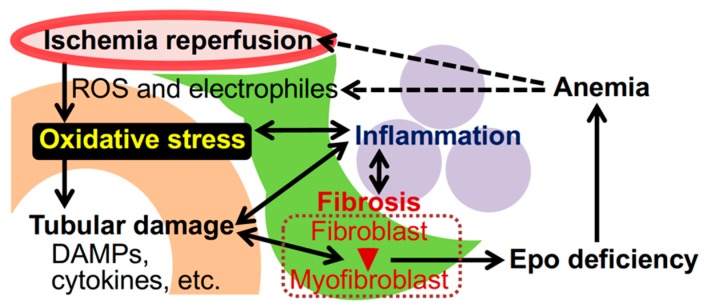
Oxidative stress and kidney disease. Irregular blood flow in the renal capillaries (red) provokes ischemia-reperfusion injury (IRI), followed by the emergence of ROS and electrophiles, which are causes of oxidative stress. Accumulated oxidative stress injures epithelial cells of renal tubules (orange), and damaged cells scatter damage-associated molecular patterns (DAMPs) and inflammatory cytokines. Tubular damage induces inflammation mediated by myeloid cells (purple) as well as transformation of interstitial fibroblasts (green) into myofibroblasts that promote kidney fibrosis. The myofibroblastic transformation impairs Epo production by renal interstitial fibroblasts (also known as REP [renal Epo-producing] cells) and causes renal anemia. Anemia and inflammation further enhance oxidative stress in kidneys.

**Figure 2 ijms-21-02951-f002:**
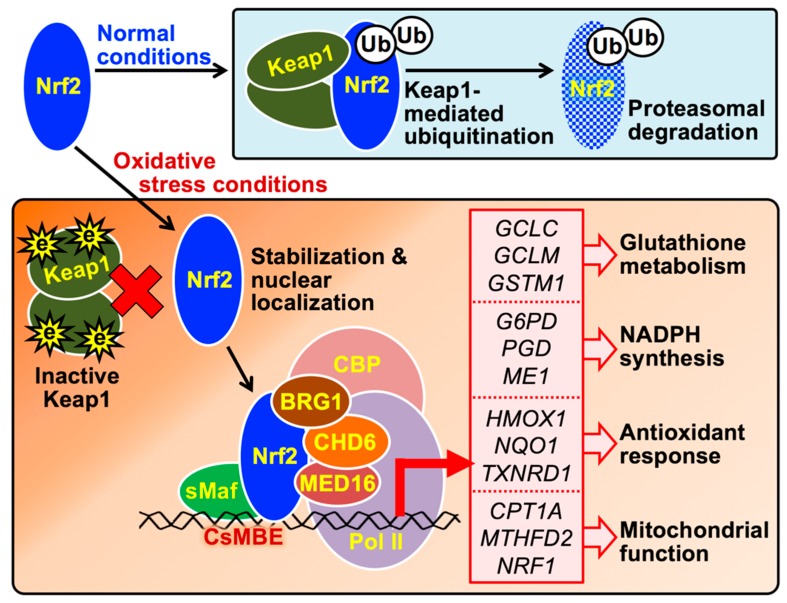
The molecular mechanism by which Nrf2 induces the expression of genes involved in cytoprotection against oxidative stress. In cells free of oxidative stress, the newly synthesized Nrf2 protein is directly captured by the Keap1 homodimer and degraded through the ubiquitin-proteasome pathway. Under oxidative stress conditions, Nrf2 avoids Keap1-mediated degradation because Keap1 is denatured by ROS or electrophiles (e). Stabilized Nrf2 is translocated into the nucleus and forms heterodimers with small Maf (sMaf) proteins, enabling binding to CsMBE in the promoters or enhancers of antioxidant genes (listed in the red box); transcription of these genes is induced by the Nrf2 complex containing CBP, BRG1, CHD6, and MED16. *HMOX1*, heme oxygenase 1; *NQO1*, NADPH quinone oxidoreductase 1; *TXNRD1*, thioredoxin reductase 1; *G6PD*, glucose-6-phosphate dehydrogenase; *PGD*, 6-phosphogluconate dehydrogenase; *ME1*, malic enzyme 1; *GCLC*, glutamate cysteine ligase catalytic subunit; *GCLM*, glutamate cysteine ligase modifier subunit; *GSTM1*, glutathione S-transferase µ1; *CPT1A*, carnitine palmitoyltransferase 1A; *MTHFD2*, methylenetetrahydrofolate dehydrogenase 2; *NRF1*, nuclear respiratory factor 1.

**Figure 3 ijms-21-02951-f003:**
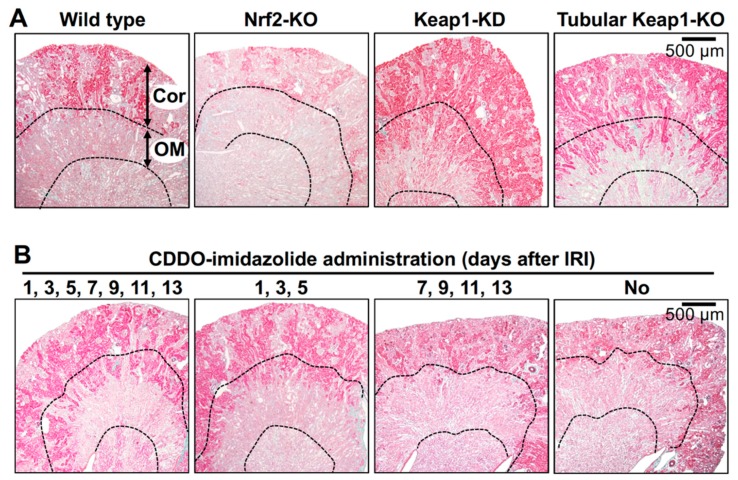
Impact of genetic and pharmacological modification of Nrf2 activity on kidney damage caused by IRI. Elastica-Masson staining of mouse kidney sections 14 days after renal IRI surgery distinguishes damaged renal tubules (pale pink) from undamaged renal tubules (deep pink) [67]. (**A**) In the outer medulla (OM), most tubules in wild-type and Nrf2-KO mice are damaged, while those in Keap1-KD and adult-onset, tubule-specific, Keap1 knockout (Tubular Keap1-KO) mice are well preserved. In the renal cortex (Cor), the damaged area is expanded in the Nrf2-KO mouse compared to that in the wild-type mouse. (**B**) The Nrf2 activator CDDO-imidazolide (30 µmol/kg body weight) was orally administered on the indicated days after renal IRI surgery, and kidney samples were obtained 14 days after the surgery. Notably, the data indicate that CDDO-imidazolide treatment in the early stages (1, 3 and 5 days) but not the later stages (7, 9, 11 and 13 days) after surgery is effective for renoprotection [67].

**Figure 4 ijms-21-02951-f004:**
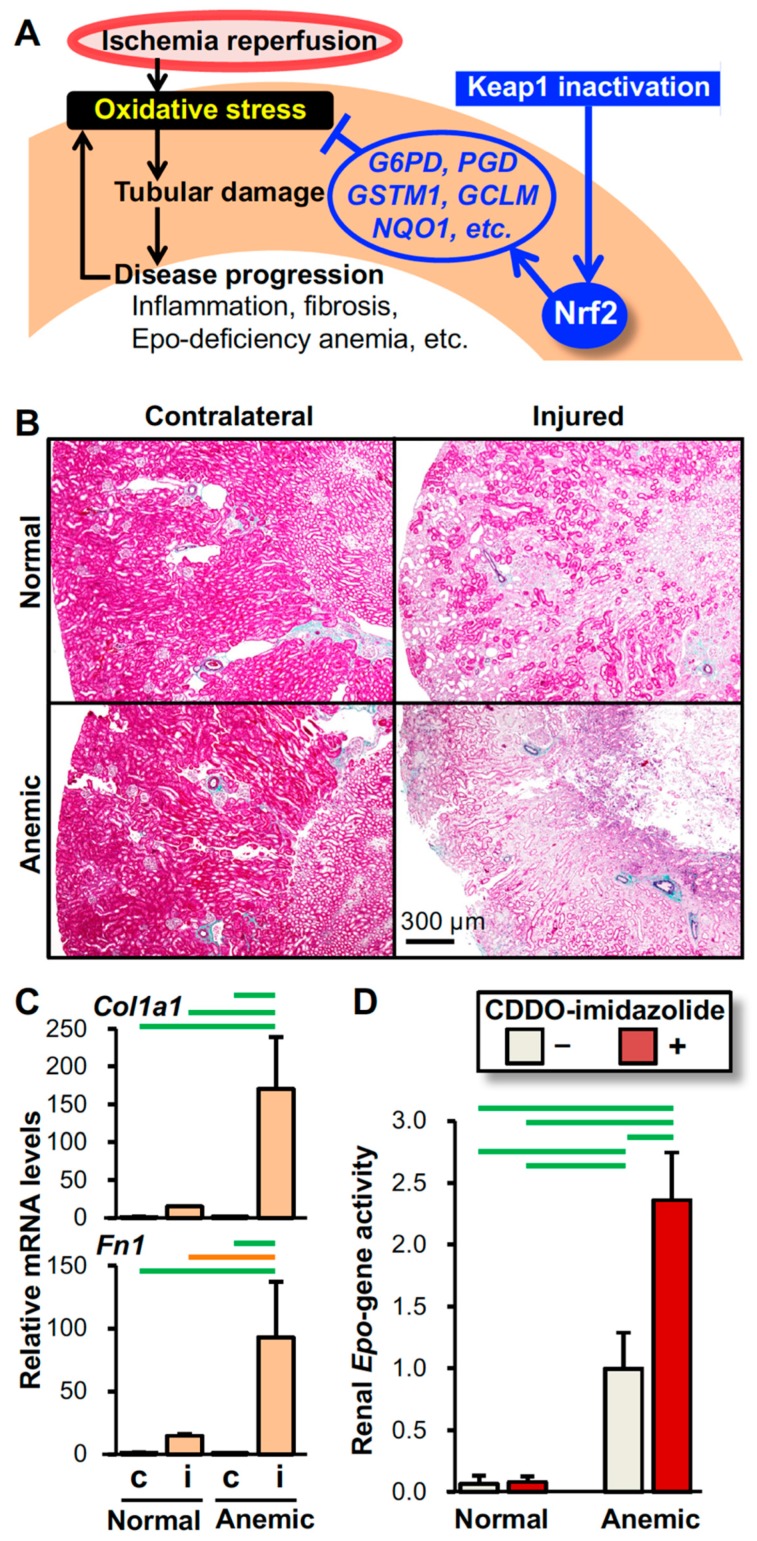
Mechanism underlying the intervention by Nrf2 activation in oxidative kidney injury. (**A**) Schematic model showing that Nrf2 protects kidneys from IRI (see Figure 1). Pharmacological inactivation of Keap1 after IRI is effective in protecting kidneys from oxidative injury by inducing the expression of antioxidative genes, which are targets of Nrf2 (see Figure 2). (**B**) Epo-deficiency anemia aggravates IRI-induced damage in kidneys. Elastica-Masson staining of the injured and contralateral kidneys from the anemic and normal mice 14 days after unilateral renal IRI demonstrates that the pale pink damaged area expands in the injured kidney of the mouse model for Epo-deficiency anemia compared to that of the normal control mouse. (**C**) Expression of the *Col1a1* and *Fn1* genes (encoding type-1α1 collagen and fibronectin, respectively) in injured (i) and contralateral (c) kidneys from anemic and normal mice 14 days after unilateral renal IRI surgery demonstrates that Epo-deficiency anemia aggravates kidney fibrosis. Notably, the expression levels of these fibrosis-related genes are very low in the uninjured contralateral kidneys. (**D**) The Nrf2 activator CDDO-imidazolide enhances renal *Epo* gene activity induced by anemic conditions. CDDO-imidazolide (30 µmol/kg body weight) was orally administered to anemic and normal mice on days 1 and 3 and then, RNA samples from kidneys were prepared on day 4. Since the *Epo* gene was replaced with GFP (green fluorescent protein) cDNA in the anemic mice, the renal *Epo* gene activity was estimated with the expression levels of the *GFP* gene. In the control mice, the *Epo* gene was heterozygously replaced with GFP cDNA [111]. *n* = 3 (C) and *n* = 5 (D) per group. The data are shown as the mean with standard deviations (C and D). *p*  <  0.01 (green line) and *p*  <  0.05 (orange line) among the groups, determined by the Tukey-Kramer highest significance difference test (C and D).

**Table 1 ijms-21-02951-t001:** Summary of mouse genetic analyses elucidating the roles of the Keap1-Nrf2 system in kidney disease.

Mouse	Phenotype in Kidney Disease	References
Nrf2-KO (systemic)	Vulnerable to kidney disease models (autoimmune nephritis, diabetic nephropathy, toxic injury, ureteral obstruction, podocyte injury, and IRI)	[56,57,58,59,60,61,62,63,64,65,66,67]
	Resistant to kidney disease models (autoimmune nephritis and diabetic nephropathy)	[71,72]
Keap1-KD (systemic)	Resistant to kidney disease models (ureteral obstruction, podocyte injury, and IRI)	[63,64,67]
Keap1-KO (myeloid cell)	No remarkable effect against renal IRI.	[67]
(T cell)	Resistant to IRI	[66]
(developing tubule)	Hydronephrosis in neonates	[69,70]
(adult tubule)	Resistant to IRI	[67]

**Table 2 ijms-21-02951-t002:** Examples of Nrf2-activating reagents experimentally tested in kidney disease models.

Compound	Kidney Disease Model	Outcome	Reference
CDDO-imidazolide	IRI (m)	Improved	[65,67]
CDDO-ME	IRI (m), toxic injury (m), and proteinuria-induced tubular damage (m)	Improved	[82,83,84]
Curcumin	Toxic injury (r, m) and 5/6 nephrectomy (r)	Improved	[85,86,87,88,89]
Dexmedetomidine	Lipopolysaccharide-induced injury (r)	Improved	[90]
Dimethyl fumarate	Toxic injury (r) and ureteral obstruction (m)	Improved	[91,92]
Farrerol	Toxic injury (r)	Improved	[93]
Omaveloxolone	IRI (m)	Improved	[94]
Paeonol	Toxic injury (m)	Improved	[95]
Resveratrol	IRI (r) and aging-related injury (m)	Improved	[96,97]
Roxadustat	Toxic injury (m)	Improved	[98]
SB216763	Toxic injury (m)	Improved	[99]
Sulforaphane	IRI (m), pristane-induced Lupus nephritis (m), and contrast-induced nephropathy (r)	Improved	[100,101,102]

m, mouse; r, rat.

**Table 3 ijms-21-02951-t003:** Clinical trials investigating CDDO-ME in diabetic CKD patients.

	Phase	Period	CKD Stage	Registration ID
BEAM	II	2009–2010	3–4	NCT00811889
BEACON	III	2011–2012 ^†^	4	NCT01351675
TSUBAKI	II	2015–2017	3–4	NCT02316821
AYAME	III	2018–2022	3–4	NCT03550443

^†^ Halted due to a high incidence of cardiovascular events.

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
