# Peer review of "Roles of Nrf2 in Protecting the Kidney from Oxidative Damage"

_ijms, 2020, doi:10.3390/ijms21082951_

Round 1

Reviewer 1 Report

The authors have made a review article on the reno-protective effects of Nrf2 and injury induced by oxidative stress. The negative regulatory role of Keap1 was also introduced. In general, the writing is good and several suggestions are listed as below:

  1. The details of how Keap1 regulate Nrf2 is totally absent in this article. Further information is required to elucidate their interactions.
  2. The authors mentioned that in tumor cells, Nrf2 can promote cell proliferation with activation of pentose phosphatase pathway and subsequent gene expression (G6pdh, Pgd). How to translate this effect into cells repairs during IRI and oxidative stress is not clarified in their manuscript (Page 5, line 175-190).
  3. The writing is incorrect in page 6, line 195-196
  4. The development and progress of Nrf2 activator (CDDO-imidazolide) is very important. I will recommend the author to make table for summarizing these results, such as BEAM, BEACON, AYAME.  

Author Response

The authors have made a review article on the reno-protective effects of Nrf2 and injury induced by oxidative stress. The negative regulatory role of Keap1 was also introduced. In general, the writing is good and several suggestions are listed as below:

We thank the reviewer for understanding the merits of this manuscript and providing professional comments.

1) The details of how Keap1 regulate Nrf2 is totally absent in this article. Further information is required to elucidate their interactions.

As both reviewers noted, in the original manuscript, we omitted to explain the details of the Keap1-Nrf2 pathway, which are repetitively introduced by the other published articles in this special issue on “The Nrf2 pathway”. Accordingly, the regulatory mechanism of Nrf2-inducible transcription in response to oxidative stress is described in the revised manuscript with a new figure and additional citations (Lines 105-124 and Figure 2).

2) The authors mentioned that in tumor cells, Nrf2 can promote cell proliferation with activation of pentose phosphatase pathway and subsequent gene expression (G6pdh, Pgd). How to translate this effect into cells repairs during IRI and oxidative stress is not clarified in their manuscript (Page 5, line 185-190).

According to the comment, we modified this important issue to clearly explain that the antioxidant NADPH produced by Nrf2-inducible PPP is suggested to play an important role in renoprotection from oxidative injury but not cell proliferation (Lines 201-210).

3) The writing is incorrect in page 6, line 195-196

We corrected this sentence (Line 213).

4) The development and progress of Nrf2 activator (CDDO-imidazolide) is very important. I will recommend the author to make table for summarizing these results, such as BEAM, BEACON, AYAME.  

A table showing an overview of clinical trials investigating Nrf2 inducers was added in the revised manuscript (Table 3).

Reviewer 2 Report

In this review authors introduce the role of Nrf2 in oxidative stress responses in kidney diseses.  This review helps to explore the better understanding of the role of Nrf2 in kidney mouse genetic models, the beneficial effects of CDDO-imidazolide, which is the most predominant Nrf2 activator. The last part of review focuses on the clinical impact of Nrf2 activator, CDDO-methyl ester on kidney injury. The main messages of this review are that Nrf2 has an important function in renoprotection and reducing oxidative stress response by Nrf2 activation is expected to mitigate a kidney disease.

In my opinion more information needs to be added to increase the comprehensiveness of this review for the reader.

Major corrections:

  1. Introduce Nrf2/Keap1 antioxidant pathway (or use schematic model)- structure, localization, signaling, NRF2 target genes. What are the main Nrf2 target genes?
  2. Figure 2 is less comprehensible in its present form, authors need to add more informations to the figure and also provide references.
  3. Authors should briefly elaborate the role of Nrf2 in human kidney cancer cells.
  4. I am not sure that Figure 3, Figure 4 B, C are appropriate (figures include primary data from research group in a review). - I recommend making a schematic image from topic with references.

If presentation of primary data is allowed, please correct the followings:

- Fig.4 B and C: statistical analysis is missing, are these results significant?

- Fig.4 C: columns marked by c are not visible, please use two segmented y axis for better visualisation.

- Abbreviations at the end of the figure legend are useful, e.g. Fig 4 : full name of G6pdx, Pgd, Gstm1, Gclm, Nqo1, Epo, GFP are completely missing from figure legend.

  1. Authors can also mention other NRF2 activators tested in a kidney disease models, besides CDDO-imidazolide, or list these activators in the table (compounds, disease models, outcome, references).
  2. Simplify part 6- Clinical impact of Nrf2 activation and/or use a table for summarizing these results with references.

Minor corrections:

  1. Line 82: Figure 1 not Figure1A
  2. Line 55: use only abbreviation IRI or remove IRI abbreviation from the abstract line 16.

Author Response

In this review authors introduce the role of Nrf2 in oxidative stress responses in kidney diseses. This review helps to explore the better understanding of the role of Nrf2 in kidney mouse genetic models, the beneficial effects of CDDO-imidazolide, which is the most predominant Nrf2 activator. The last part of review focuses on the clinical impact of Nrf2 activator, CDDO-methyl ester on kidney injury. The main messages of this review are that Nrf2 has an important function in renoprotection and reducing oxidative stress response by Nrf2 activation is expected to mitigate a kidney disease.  In my opinion more information needs to be added to increase the comprehensiveness of this review for the reader.

We appreciate the reviewer for these professional and constructive comments.

1) Introduce Nrf2/Keap1 antioxidant pathway (or use schematic model)- structure, localization, signaling, NRF2 target genes. What are the main Nrf2 target genes?

According to the reviewer’s suggestion, we added a section (Lines 105-124) and a figure (Figure 2), both of which describe the regulatory mechanism of the Nrf2-mediated transcription response to oxidative stress.

2) Figure 2 is less comprehensible in its present form, authors need to add more informations to the figure and also provide references.

The figure (Figure 2 in the original manuscript) was modified and transformed into a table (Table 1) containing detailed information regarding the renal phenotype of mice bearing genetic modifications in the Keap1-Nrf2 system. Appropriate citations were also added to the table.

3) Authors should briefly elaborate the role of Nrf2 in human kidney cancer cells.

Accordingly, we described the putative roles of Nrf2 in kidney cancer in the revised manuscript. Additionally, we discussed the negative effects of Nrf2 activators in treating kidney cancer and kidney disease (Lines 245-252).

4) I am not sure that Figure 3, Figure 4 B, C are appropriate (figures include primary data from research group in a review). - I recommend making a schematic image from topic with references.  If presentation of primary data is allowed, please correct the followings:

- Fig.4 B and C: statistical analysis is missing, are these results significant?

- Fig.4 C: columns marked by c are not visible, please use two segmented y axis for better visualisation.

- Abbreviations at the end of the figure legend are useful, e.g. Fig 4 : full name of G6pdx, Pgd, Gstm1, Gclm, Nqo1, Epo, GFP are completely missing from figure legend.

As many review articles similarly contain primary data, we believe that the primary data are acceptable and more helpful for the comprehension of this review paper than schematic images.

According to the reviewer’s suggestion, we added the statistical analysis in Figure 4CD. The data demonstrate that the expression levels of fibrosis-related genes in injured kidneys significantly differ between normal and anemic mice (Figure 4C) and that CDDO-imidazolide significantly enhances renal Epo-gene activity in anemic mice (Figure 4D).

However, we did not segment the y-axes in Figure 4C because the expression levels of genes related to fibrosis are known to be very low in non-injured control kidneys (“c” in Figure 4C). We used the data as negative controls. This issue is explained in the legend of Figure 4C in the revised manuscript (Line 291).

The abbreviations of the gene names were added to Figure 2.

5) Authors can also mention other NRF2 activators tested in a kidney disease models, besides CDDO-imidazolide, or list these activators in the table (compounds, disease models, outcome, references).

A table listing Nrf2-activating compounds tested in kidney disease models was added to the revised manuscript (Table 2).

6) Simplify part 6- Clinical impact of Nrf2 activation and/or use a table for summarizing these results with references.

We summarized the clinical trials investigating CDDO-ME for the treatment of diabetic CKD in Table 3.

Minor corrections:

Line 82: Figure 1 not Figure1A

Line 55: use only abbreviation IRI or remove IRI abbreviation from the abstract line 16.

These issues were corrected.

Round 2

Reviewer 2 Report

This review summarizes the roles of Nrf2 in protecting the kidney from oxidative stress.  

In the revised version authors added more information to the review (figures, tables, references) to increase the comprehensiveness of the review for the reader.

  • Authors described the regulatory mechanism of the Nrf2 antioxidant pathway to oxidative stress and added Figure 2 (new) which summarizes these infomations.
  • Figure 2 from the original article was removed and informations were transfered into table (Table 1)- summary of mouse genetic analyses elucidating the roles of the Nrf2 antioxidant system in kidney disease.
  • Authors added Table 2 (new)- list of Nrf2 activating compounds tested and also discussed the negative effect of Nrf2 activators in kidney diseases.
  • Table 3 (new) summarizes the clinical trials investigating CDDO-ME in diabetic CKD patients.

The authors made all the requested modifications in the review.

I recommend to accept this review in its present form.